# Evidence for a role of human blood-borne factors in mediating age-associated changes in molecular circadian rhythms

Jessica E Schwarz[1,2], Antonijo Mrčela[3], Nicholas F Lahens[3], Yongjun Li[2], Cynthia Hsu[1,2], Gregory R Grant[3,4], Carsten Skarke[2,3], Shirley L Zhang[1,2]*†, Amita Sehgal[1,2]*

[1]Howard Hughes Medical Institute, Perelman School of Medicine, University of Pennsylvania, Philadelphia, United States; [2]Chronobiology and Sleep Institute, Perelman School of Medicine, University of Pennsylvania, Philadelphia, United States; [3]Institute for Translational Medicine and Therapeutics (ITMAT), Perelman School of Medicine, University of Pennsylvania, Philadelphia, United States; [4]Department of Genetics, Perelman School of Medicine, University of Pennsylvania, Philadelphia, United States

*For correspondence:
shirley.zhang2@emory.edu (SLZ);
amita@pennmedicine.upenn.
edu (AS)

Present address: †Department of Cell Biology, Emory University School of Medicine, Atlanta, United States

## eLife Assessment

The authors tested the hypothesis that age-dependent factors in human sera affect the core circadian clock or its outputs in cultured fibroblasts, and they provide **compelling** evidence that genes involved in the cell cycle and transcription/translation remain rhythmic in both conditions, genes associated with oxidative phosphorylation and Alzheimer's Disease lose rhythmicity in the aged condition, while the expression of cycling genes associated with cholesterol biosynthesis increase in the cells entrained with old serum. Together, the findings suggest that yet to be identified age-dependent blood-borne factors affect circadian rhythms in the periphery. The paper provides **fundamental** insights and a possible explanation for previous observations showing that circadian gene expression in peripheral tissues tends to dampen or phase-shift with age.

**Abstract** Aging is associated with a number of physiologic changes including perturbed circadian rhythms; however, mechanisms by which rhythms are altered remain unknown. To test the idea that circulating factors mediate age-dependent changes in peripheral rhythms, we compared the ability of human serum from young and old individuals to synchronize circadian rhythms in culture. We collected blood from apparently healthy young (age 25–30) and old (age 70–76) individuals at 14:00 and used the serum to synchronize cultured fibroblasts. We found that young and old sera are equally competent at initiating robust ~24 hr oscillations of a luciferase reporter driven by clock gene promoter. However, cyclic gene expression is affected, such that young and old sera promote cycling of different sets of genes. Genes that lose rhythmicity with old serum entrainment are associated with oxidative phosphorylation and Alzheimer's Disease as identified by STRING and IPA analyses. Conversely, the expression of cycling genes associated with cholesterol biosynthesis increased in the cells entrained with old serum. Genes involved in the cell cycle and transcription/translation remain rhythmic in both conditions. We did not observe a global difference in the distribution of phase between groups, but found that peak expression of several clock-controlled genes (*PER3, NR1D1, NR1D2, CRY1, CRY2,* and *TEF*) lagged in the cells synchronized ex vivo with old serum. Taken together, these findings demonstrate that age-dependent blood-borne factors affect circadian rhythms in peripheral cells and have the potential to impact health and disease via maintaining or disrupting rhythms respectively.

## Introduction

Circadian rhythms are known to regulate homeostatic physiology including sleep:wake, hormone production and body temperature, and their dysregulation with aging is accompanied by adverse health consequences (*Monk et al., 1993*; *Monk et al., 2000*; *Hood and Amir, 2017*), raising the possibility that health decline with age is caused in part by circadian dysfunction. Although the mechanisms responsible for age effects on circadian rhythms are unknown, signals from the central clock in the suprachiasmatic nucleus (SCN) dampen with age (*Hood and Amir, 2017*; *Farajnia et al., 2012*; *Nakamura et al., 2011*) and rhythms change in peripheral tissues in different ways (*Nakamura et al., 2011*; *Yamazaki et al., 2002*). Here, we aimed to develop a cell culture model to study the effect of aging on human rhythms of peripheral tissues. Given that serum can reset the clock in peripheral fibroblasts (*Gerber et al., 2013*), we questioned the extent to which serum factors normally contribute to peripheral rhythms of gene expression and how they might affect rhythms with age, given that blood-borne factors can influence other aspects of aging (*Katsimpardi et al., 2014*).

Using an established clinical study paradigm that sampled blood at 14:00 (*Pagani et al., 2011*; *Qin et al., 2020*), we collected blood from young (age 25–30) and old (age 70–76) apparently healthy individuals with behavioral and physiological outputs quantified by wearable devices, and we tested the hypothesis that age-dependent factors in the sera affect circadian rhythms in cultured fibroblasts. In support of this hypothesis, we show here that genes associated with oxidative phosphorylation and mitochondrial functions lose rhythmicity in fibroblasts exposed to aged serum factors. We also find evidence of reduced entrainment in terms of altered expression of several molecular clock genes (*PER3, NR1D1, NR1D2, CRY1, CRY2*, and *TEF*) when synchronized with aged serum. These findings suggest that age-related changes in blood borne factors contribute to impaired circadian physiology and the associated disease risks.

## Results

We enrolled eight old and seven young human subjects (*Figure 1A*, *Figure 1—figure supplement 1*), whose demographics are listed in *Supplementary file 1*. Behavioral and physiological assessments confirmed entrainment to the light-dark conditions local to the East Coast of the US. This is, for example, evident in the diurnal rhythms observed for physical activity (*Figure 1C* top, vector magnitude two-way ANOVA for time-of-day $q$=3.3E-06), light exposure (lux two-way ANOVA for time-of-day $q$=0.01), heart rate (bpm two-way ANOVA for time-of-day $q$=0.016) and sympathetic and parasympathetic nervous system indices derived from the Kubios heart rate variability analysis (two-way ANOVA for time-of-day $q$=0.099 and $q$=0.063, respectively). Notably BioPatch EKG data were not obtained from all subjects (*Supplementary file 2*).

Age-specific differences emerged for several outputs. The standard deviation of instantaneous heart rate values was significantly lower in the old compared to young (two-way ANOVA for age group q=0.042). As expected, the peak clock time (acrophase) of physical activity (triaxial accelerometry integrated as vector magnitude) was phase-advanced among old compared to young subjects, as was the midsleep time calculated from the MCTQ (*Figure 1—figure supplement 2B*), though not at a statistically significant level (p=0.055). On average, a lower rhythm-adjusted mean (mesor) heart rate of 55.8±3.5 bpm was found in old compared to 69.7±5.0 bmp in young, along with a higher heart rate variability (RR intervals) of 1106.7±80.5ms compared to 893.2±61.8ms, respectively (*Figure 1B*). Old subjects displayed lower activity in the sympathetic nervous system (SNS), and higher activity in the parasympathetic nervous system (PNS) compared to young (*Figure 1—figure supplement 2A*), with no significant difference in cortisol levels (*Figure 1—figure supplement 2C*). The overall high degree of variability rendered the cardiovascular trends, however, statistically not significant.

As noted above, blood was collected from these old/young individuals and the serum was used to synchronize BJ-5TA fibroblasts stably transfected with a *BMAL1- luciferase* construct (*Gerber et al., 2013*; *Pagani et al., 2011*; *Ramanathan et al., 2012*; *Balsalobre et al., 1998*). Circadian rhythms were assessed by luciferase assay (*Ramanathan et al., 2012*) over 4 days (*Figure 2A*). We conducted this study in a fibroblast line in order to build upon age-related changes found in a previous study which used human fibroblast lines (*Pagani et al., 2011*). An additional rationale for using this line is that it derives from normal human tissue, providing an advantage in studying normal physiology when compared to the commonly used U2OS line which is a genomically unstable osteosarcoma with

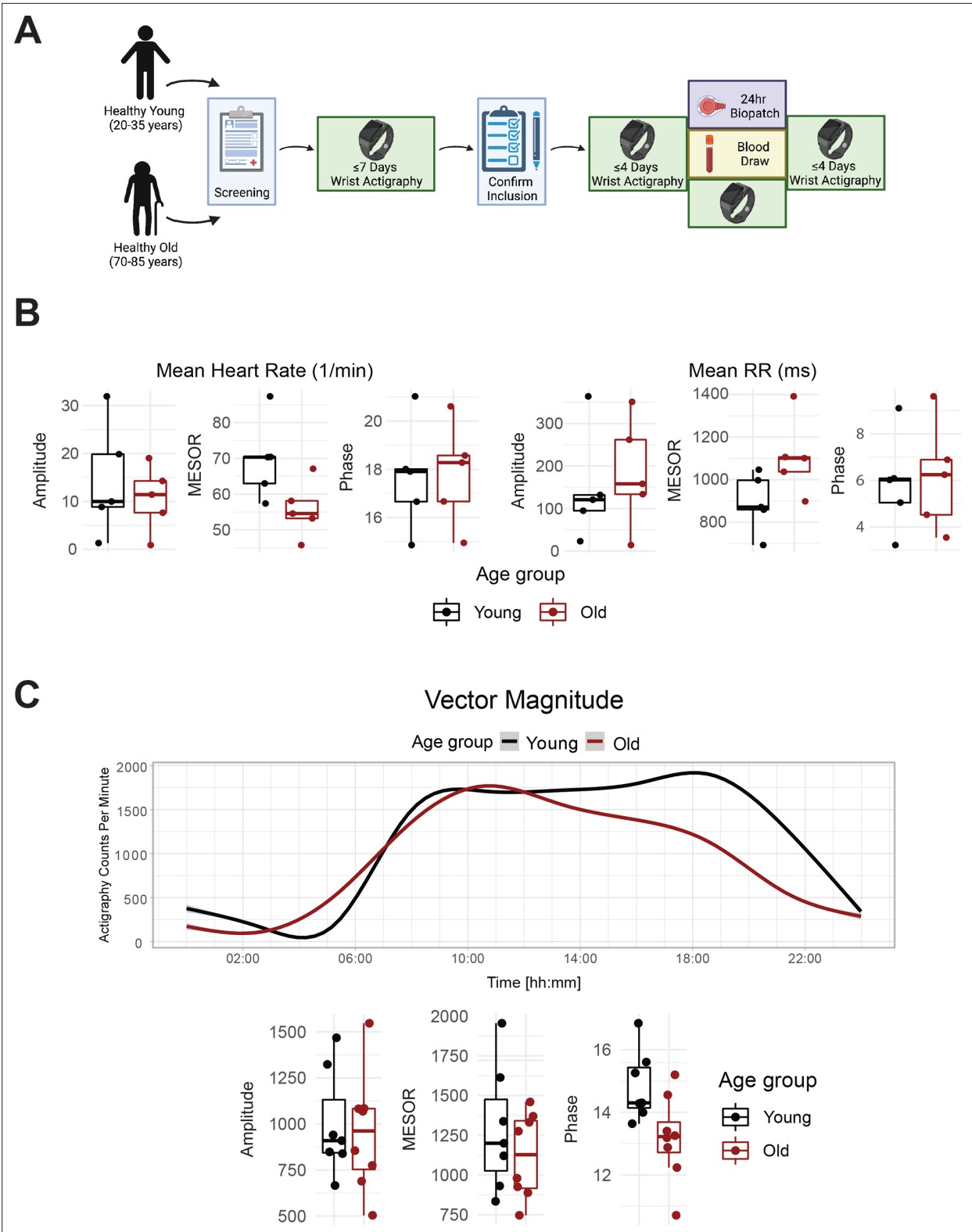

**Figure 1.** Healthy elderly individuals tend to have lower heart rate, increased heart rate variability, and phase advanced activity patterns relative to young subjects. Experimental protocol for enrollment and monitoring of human subjects (**A**). As assessed by Zyphyr BioPatch, electrocardiogram (EKG) measurements suggest that MESOR of average heart rate decreases with age (p=0.056) and that MESOR of heart rate variability increases with age (p=0.056). N=5 per group. MESOR and amplitude were tested using two-sided Wilcoxon rank sum exact test, while phase was tested by Kuiper's two-sample test (**B**). Average activity counts across three axes (vector magnitude), as recorded by the actigraph device plotted throughout the day (top)

*Figure 1 continued on next page*

*Figure 1 continued*

and analyzed for circadian rhythm (bottom) (**C**). While the amplitudes of activity did not differ between the age groups, the older individuals trended towards an early phase (p~0.055, Kupier's two-sample test) compared to young individuals. N=7 for young and N=8 for old. Lines in the top panel of C are smoothed means (fit with penalized cubic regression splines) for data from each age group. Dots in the bottom panel of C are subject-level cosinor parameter estimates derived from cosinor fits to the actigraphy data. Boxplot midlines correspond to median values, while the lower and upper hinges correspond to the first and their quartiles, respectively. Boxplot whiskers extend to the smallest/largest points within 1.5 * IQR (Inter Quartile Range) of the lower/upper hinge. Panel A was created with BioRender.com.

The online version of this article includes the following source data and figure supplement(s) for figure 1:

**Source data 1.** A flow chart of the number of subjects included in each analysis present in this study.

**Figure supplement 1.** Subject inclusion flow chart.

**Figure supplement 2.** There are differences in midsleep time, but no significant differences in sympathetic, parapsympathetic nervous system indices, or cortisol levels between young and old individuals.

**Figure supplement 2—source data 1.** Raw data obtained from serum cortisol measurements.

chromosomal abnormalities (*Al-Romaih et al., 2003*). No significant differences were observed in the period, amplitude, and phase of the *BMAL1-luciferase* rhythm with young versus old serum treatment (*Figure 2—figure supplement 1*).

To assess serum effects on circadian gene expression, we first performed RNA-seq on fibroblasts synchronized with serum from a single old or single young individual at two hour intervals and found that, compared to the first day of synchronization, the second day (36–58 hr after synchronization) showed greater differences in MESORs between young and old serum-treated groups (*Figure 2— figure supplement 2*). This is not surprising because the first day includes acute responses of the fibroblasts to serum, which can mask circadian rhythms, and so the second day is expected to reveal differences in endogenous rhythms between samples. Day 2 also revealed different phases of cyclic expression between young and old subjects for a larger number of genes. We proceeded to collect fibroblasts synchronized by sera from eight different subjects (four old, four young with two male and two female per group) at 2-hr intervals, from 32 to 58 hr post serum addition. This added an extra two timepoints to the second day to facilitate the calculation of rhythmicity. Using CircaCompare *Parsons et al., 2020* and a weighted Bayesian Information Criterion (BIC)>0.75 cutoff, a significant number of genes were found to lose (1519 genes) or gain (637 genes) rhythmicity with age, underscoring the impact of age on the ability of serum to synchronize circadian rhythms, while 1209 genes were rhythmic in both groups (*Figures 2B and 3A*). Additionally, we used CircaCompare to estimate MESORs, amplitudes, and phases of gene oscillatory patterns in young and old groups, and to compare these cosinor parameters between groups (*Figure 2C–D*). Of the genes that were rhythmic in both groups, many also showed changes with age. For instance, 568 cyclically expressed genes showed a change in MESOR with age (q<0.05 for MESOR differences), with MESOR values increasing for 163 genes and decreasing for 405 genes in the old serum-treated samples (*Figure 2C*, *Figure 2—figure supplement 3*, right). Using q-values provided by CircaCompare we were able to detect changes in amplitude (*Figure 2D*, *Figure 2—figure supplement 3*, left) for only a small number of genes (39 genes had decreased amplitude in old and 2 had increased amplitude). Using CircaCompare and a q<0.05 cutoff for phase differences in genes rhythmic in young and old, we detected 20 genes with advanced phase, and 34 with delayed phases (*Figure 2E*). However, it is important to note that in these analyses low p-values can be driven by large sample-size such as in *Figure 2—figure supplement 3* where distributions of MESORs and amplitudes across genes are assessed such that each gene represents a sample.

As mentioned, we found several examples of transcripts that cycle with young serum synchronization and lose rhythmicity (ex. TCF4), decrease in amplitude (ex. HMGB2), phase shift (ex. TEF), or change MESOR (ex. HSD17B7) with aged serum (*Figure 2—figure supplement 4*). Search Tool for the Retrieval of Interacting Genes (STRING) *Szklarczyk et al., 2019* and Ingenuity Pathway Analysis (IPA) *Krämer et al., 2014* were used for functional genomics. Both approaches indicate a maintenance of cycling of cell cycle genes with young and old serum synchronization, and a loss of rhythmicity of genes associated with oxidative phosphorylation in the aged serum (*Figure 3A* and *Figure 3—figure supplement 1*). STRING analysis revealed that the dominant pathways associated with genes rhythmic in both young and old conditions, cell cycle and DNA replication, demonstrate a decrease in MESOR with old serum. For instance, checkpoint control and chromosomal replication pathway associated

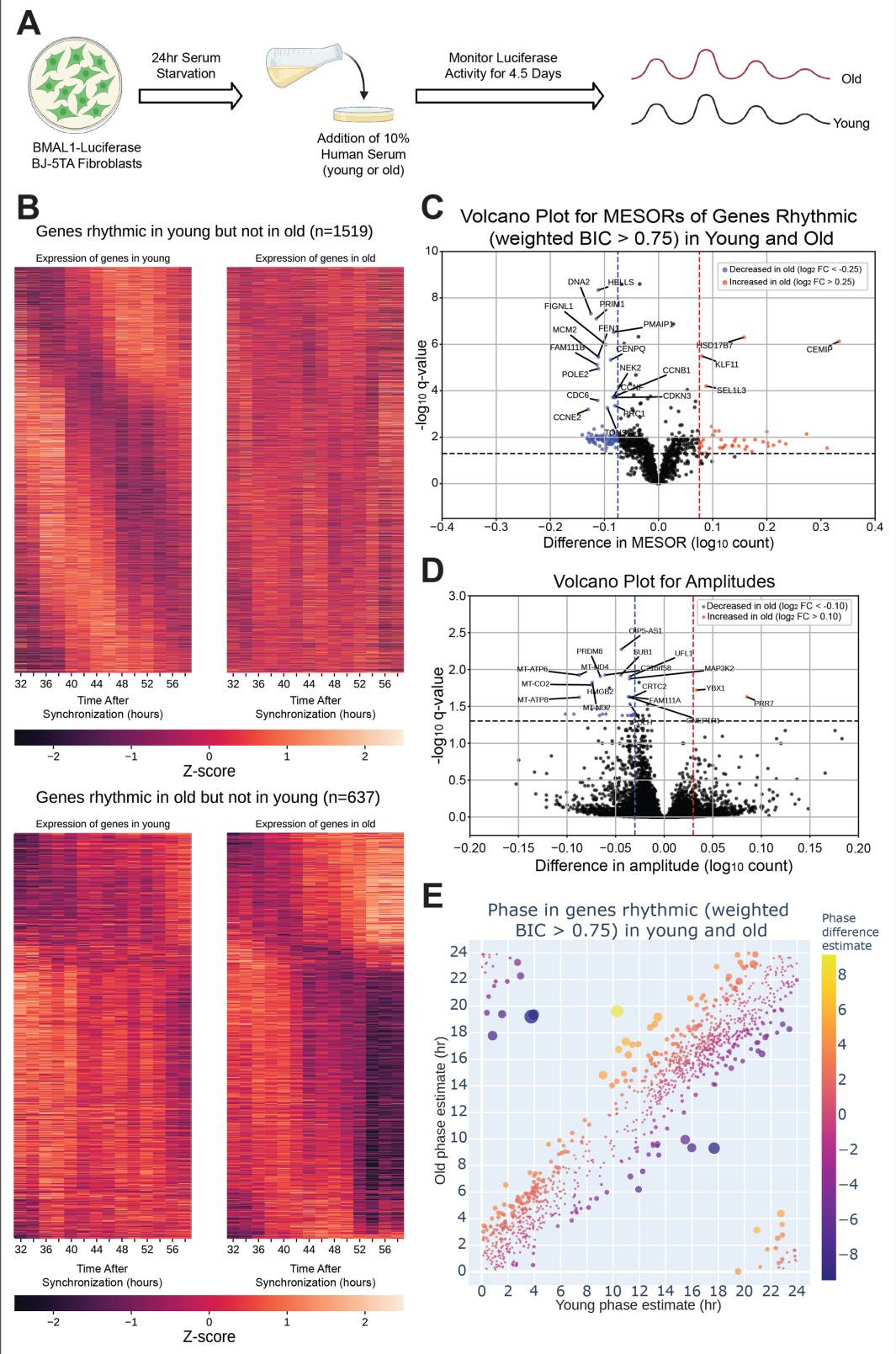

**Figure 2.** The circadian transcriptome is differentially affected by entrainment with sera from young and old human subjects. A visual representation of the serum starvation-serum addition protocol to synchronize the *BMAL1-luciferase* BJ-5TA fibroblasts (**A**). When comparing the circadian transcriptome entrained by either young or old sera (n=4 sera per group), 1519 genes lost rhythmicity with age (B, top) while only 637 genes gained rhythmicity

*Figure 2 continued on next page*

*Figure 2 continued*

with age (B, bottom). Weighted BIC criterion with a threshold of 0.75 was used to assess rhythmicity (**B**). The number of genes rhythmic in young and old (according to weighted BIC > 0.75 criterion) that show a detectable change in MESOR (q<0.05 criterion for MESOR difference) is 568, with MESOR increasing in 163 and decreasing in 405 genes. Out of these, 40 genes with increased MESOR (red) and 98 genes with decreased MESOR (blue) also satisfy the condition $|\log_2 FC| > 0.25$ (**C**). We were only able to detect change in amplitude for a small number of genes, 39 genes had decreased amplitude in old and 2 had increased amplitude using CircaCompare. Only 30 genes with decreased amplitude also satisfy the condition $|\log_2 FC| > 0.1$ (blue) (**D**). For phase, using a test provided by CircaCompare, under q<0.05 cutoff for age-related phase differences in genes rhythmic in young and old (with BIC > 0.75 cycling criteria), we detected 20 genes with advanced phase, and 34 with delayed phase (**E**). Panel A was created with BioRender.com.

The online version of this article includes the following source data and figure supplement(s) for figure 2:

**Figure supplement 1.** Young and old serum are equally effective at entraining cells in culture.

**Figure supplement 1—source data 1.** Circadian transcripome analysis of Day 2 of serum entrainment compared to Day 1.

**Figure supplement 2.** The circadian transcriptomes of cells synchronized with young or old sera deviate significantly on Day 2.

**Figure supplement 3.** There is a small effect of age on the amplitude and MESOR of cycling mRNA between young and old serum samples.

**Figure supplement 4.** Sample traces of transcripts that cycle with young serum synchronization and are differentially affected by old serum synchronization.

---

genes were expressed cyclically in both young and old conditions; however, several chromosomal replication pathway genes exhibit decreased MESORs in the aged sera (*Figure 3—figure supplement 1*, *Supplementary file 3*). MESORs of steroid biosynthesis genes, particularly those relating to cholesterol biosynthesis, were increased in the old sera condition (*Figure 3A*, *Supplementary file 4*).

Loss of rhythms in oxidative phosphorylation suggests mitochondrial dysfunction with age. By STRING analysis, 24 out of the 26 genes associated with oxidative phosphorylation overlap with the Alzheimer's Disease KEGG pathway highlighting the disease relevance of this class of genes that loses cycling in older individuals. Given that Alzheimer's pathology is closely associated with the accumulation of oxidative stress (*Holubiec et al., 2022*), it is possible that loss of cycling contributes to oxidative damage. An additional 31 genes in the Alzheimer's Disease KEGG pathway lose rhythmicity with aged serum; these include amyloid precursor protein (APP) and apolipoprotein E (APOE). APP is by definition the precursor of amyloid beta, which accumulates in AD (*Sehar et al., 2022*). While APOE plays an important role in lipid transport, specific variants of this gene are strongly associated with AD risk as APOE interacts with amyloid beta in amyloid plaques, a hallmark of the disease (*Raulin et al., 2022*).

To determine whether aged serum modifies the activity of specific transcription factors, we performed an epigenetic Landscape In Silico deletion analysis (LISA), a computational tool designed to predict changes in transcription factor activity based on gene expression. Using a comprehensive database of known transcription factor binding profiles of fibroblasts, we identified altered gene expression corresponding to 59 total transcription factors (q<0.05) in our young-versus-old cycling dataset. Potential changes in the activity of these transcription factors are associated with the following five categories of cycling phenotypes: decreased MESOR, increased MESOR, phase delay, gain of rhythmicity, and loss of rhythmicity, as were defined above (*Figure 3B*). Twenty-six transcription factors were represented in all groups, and included those implicated in diseases such as cancer (MYC, TP53, YAP1, RBL2, E2F7, BRD4, MXl1), inflammation (CEBPB, BHLHE40), oxidative stress (NFE2L2), and neurological conditions (NEUROG2, SUMO2).

Lastly, several clock genes showed differences in expression with the aged serum, most notably genes in the Circadian Rhythm KEGG pathway (*Figure 4*). In particular, expression of *CRY1*, *CRY2*, *NR1D1*, *NR1D2*, *PER3*, and *TEF* was significantly phase delayed after synchronization with old serum compared to young (*Figure 4*). On the other hand, although several genes in the eIF2 signaling pathway decreased cycling with age, components expressed cyclically with older serum showed a phase advance (*Figure 3—figure supplement 1* and *Figure 4—figure supplement 1*). Importantly, the RNA-seq did not reveal a difference in the phase or amplitude of *BMAL1* expression with age,

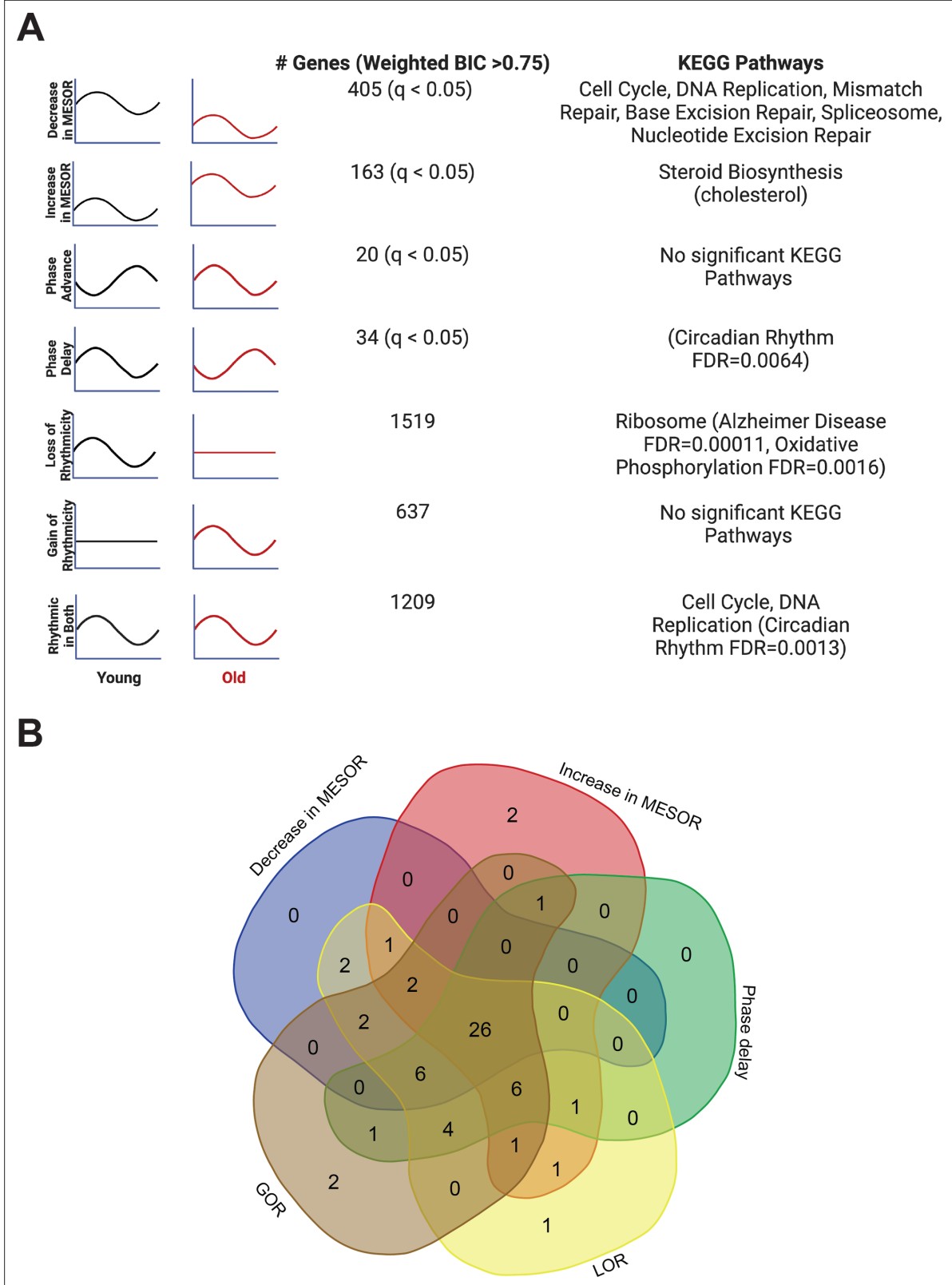

**Figure 3.** Each type of circadian change is associated with different KEGG pathways by STRING analysis, but a similar set of transcription factors identified by LISA. Entrainment of the fibroblasts in culture with old serum significantly altered the circadian transcriptome despite use of the same cells in both young and old conditions, suggesting the aged serum itself affects the regulation of specific pathways in the cell. Significant KEGG Pathways (FDR < 1 × 10⁻⁴ unless specified) are indicated next to each category of genes. Age related pathways such as Alzheimer's Disease/oxidative

*Figure 3 continued*

phosphorylation are associated with a loss of rhythms in the old condition. Cell cycle and DNA replication pathways remain rhythmic in the old serum condition, but cycle with a decreased MESOR. Rhythmic genes were determined using CircaCompare. Rhythmicity was determined using the weighted BIC > 0.75 criterion while p-values for difference in MESOR and phase were determined using CircaCompare (**A**). LinC similarity analysis (LISA), based on known transcription factor binding in fibroblasts and RNA expression, suggests that 59 total transcription factors (q<0.05) show significant changes in activity in conjunction with the following cycling phenotypes: decreased in MESOR, increased in MESOR, phase delay, gain of rhythmicity, and loss of rhythmicity as were defined above (**B**). Panel A was created with BioRender.com.

The online version of this article includes the following source data and figure supplement(s) for figure 3:

**Source data 1.** Raw data from LISA transcription factor analysis based on RNA sequencing results.

**Source data 2.** IPA analysis of oxidative phosphorylation genes and chromosomal replication genes that are affected by entrainment with old serum.

**Figure supplement 1.** Oxidative phosphorylation genes and chromosomal replication genes are affected by entrainment with old serum.

**Figure supplement 1—source data 1.** Raw output from IPA analysis of genes whose transcripts cycle in at least one condition.

supporting the validity of our *BMAL1-luciferase* findings, although the MESOR significantly increased with age.

## Discussion

Studies of the aged SCN have revealed persistent cycling of clock gene expression but a breakdown of output signals (*Farajnia et al., 2012*; *Nakamura et al., 2011*). However, whether age-related changes in systemic signaling impact tissue/organ clocks has yet to be elucidated. We demonstrate here that while the core clock continues to cycle in cultured fibroblasts synchronized with serum from old volunteers, the circadian transcriptome is different from that seen in cells treated with serum from young individuals. Through this analysis of the role of serum in age-induced changes in circadian rhythms, we suggest a potential mechanism for observations where specific genes showed a loss, gain, or maintenance of rhythmicity with age (*Chen et al., 2016*; *Kuintzle et al., 2017*; *Sato et al., 2017*; *Solanas et al., 2017*; *Blacher et al., 2022*). This phenomenon, known as circadian reprogramming, may illuminate which pathways are affected by or become more impactful for changes in cellular integrity through aging (*Wolff et al., 2023*).

In order to study the effect of circulating factors on age-related changes of the circadian transcriptome, we utilized the well-established serum starvation-serum addition protocol (*Gerber et al., 2013*) to synchronize cells in culture. Cultured fibroblasts from old and young subjects have robust clocks and respond similarly to synchronization with dexamethasone (*Pagani et al., 2011*); however, when synchronized with dexamethasone in a media containing old serum, they reportedly exhibited shortened circadian periods (*Pagani et al., 2011*). This effect was reversed by heat inactivating old serum. We did not observe a significant difference in period length between young and old serum synchronization in our *BMAL1-luciferase* experiment, perhaps because dexamethasone was not utilized. Our use of serum to synchronize allowed us to more closely simulate in vivo conditions where blood is an important mode by which central clocks can entrain peripheral clocks to maintain synchrony with the day:night cycle. Previously, serum factors were shown to activate the immediate early transcription factor, SRF, in a diurnal manner to confer time of day signals to both cells in culture and to mouse liver (*Gerber et al., 2013*). We show here that effects of age are also mediated by serum.

We find that while the number of rhythmic transcripts (weighted BIC > 0.75) in the young serum condition (~18%) is higher than the old serum condition (~12%), both conditions demonstrate a much larger number of cycling transcripts in these cultured fibroblasts than in other cell culture studies (*Duffield et al., 2002*; *Grundschober et al., 2001*; *Hughes et al., 2009*; *Jang et al., 2015*). In mammals, up to 20% of transcripts can cycle in a given tissue (*Patel et al., 2012*; *Zhang et al., 2014*), and the low number of cycling transcripts in culture has been cited as a major limitation of the culture model (*Hughes et al., 2009*). While our analysis used a normal human fibroblast cell line (BJ-5TA) we had higher statistical power, given that we had four different biological replicates at every 2 hr timepoint. However, a major contributing factor to the high rhythmicity might be the use of human serum as the synchronization signal. In this way, our model of mimicking signaling to peripheral tissues by using serum directly from humans may more accurately recapitulate the human condition. In this study, we focused only on the effects of serum synchronization on fibroblasts, but it is likely that

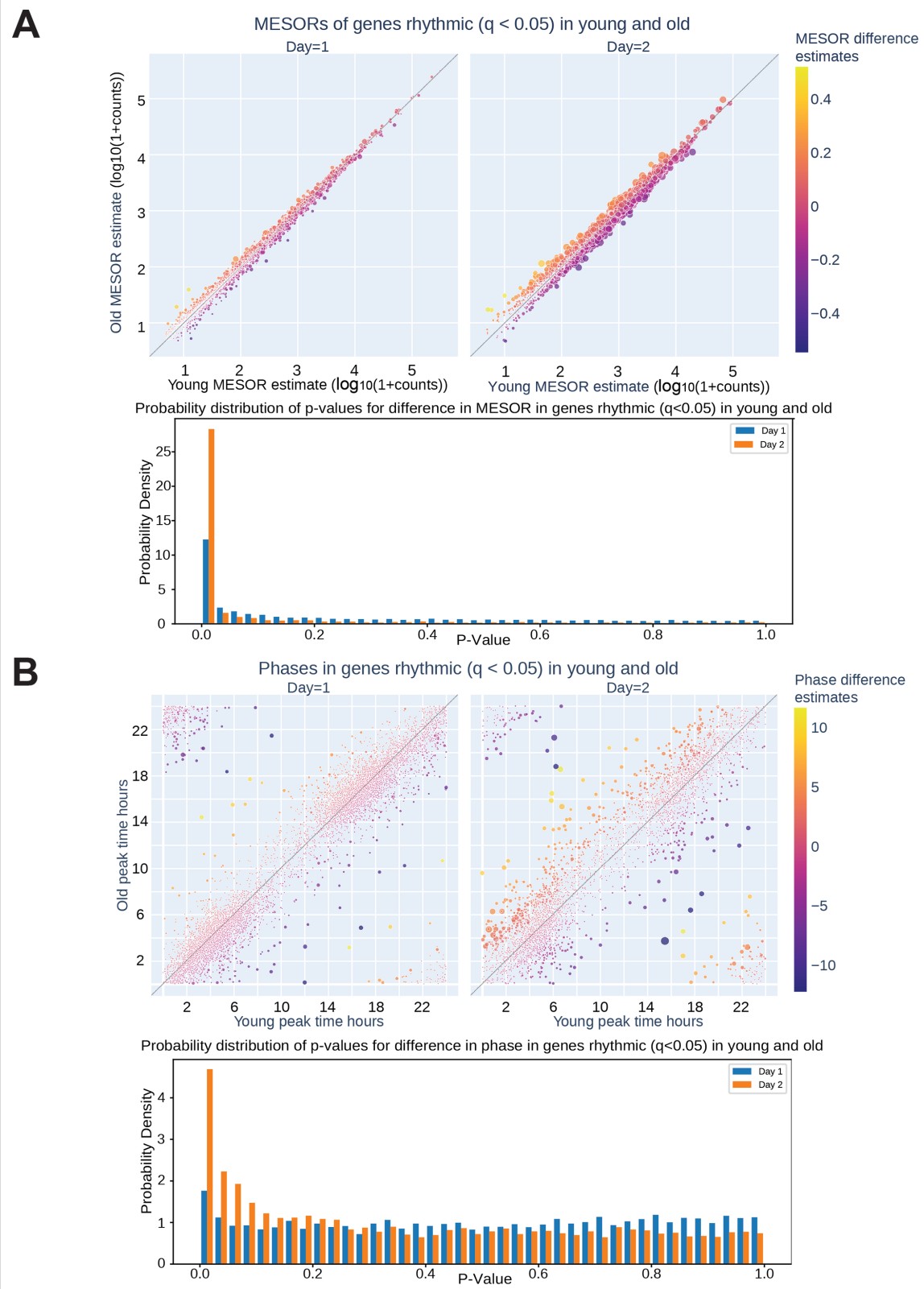

**Figure 4.** Synchronizing with old serum phase delays the expression profile of several core clock genes. Traces of molecular clock mRNA transcripts, the average curve (bold) of results of individual sera (faded). p-Values for the difference in MESOR (P_ΔM), amplitude (P_ΔA), and phase (P_ΔP) are shown for the comparison of young and old conditions. Several clock genes are significantly phase delayed (*CRY1, CRY2, NR1D1, NR1D2, PER3, TEF*) in

*Figure 4 continued on next page*

*Figure 4 continued*

response to synchronization with old serum. While *BMAL1* is not phase delayed, the MESOR significantly increases with age. N=4 subjects per timepoint for both young and old groups.

The online version of this article includes the following figure supplement(s) for figure 4:

**Figure supplement 1.** eIF2 signaling pathway genes that maintain cycling with age phase advance in the old serum condition.

factors circulating in serum act on several tissues, and so their effects are relatively broad. However, we acknowledge that in order to support this claim future studies should investigate other peripheral tissue cell types. Additionally, in the future we intend to analyze the serum using a combination of fractionation and either proteomics or metabolomics to identify relevant factors for the regulation of peripheral rhythms.

Interestingly, many of the genes in the circadian transcriptome that exhibit age-related changes are independently implicated in aging. For instance, some of the genes that lose rhythmicity in the aged condition are involved in oxidative phosphorylation and mitochondrial function, both of which decrease with age (*Lesnefsky and Hoppel, 2006*). Previous work in the field demonstrates that synchronization of the circadian clock in culture results in cycling of mitochondrial respiratory activity (*Cela et al., 2016*; *Scrima et al., 2020*) further underscoring the different effects of old serum, which does not support oscillations of oxidative phosphorylation associated transcripts. Age-dependent decrease in oxidative phosphorylation and increase in mitochondrial dysfunction (*Lesnefsky and Hoppel, 2006*) is seen also in aged fibroblasts (*Greco et al., 2003*) and contributes to age-related diseases (*Federico et al., 2012*). We suggest that the age-related inefficiency of oxidative phosphorylation is conferred by serum signals to the cells such that oxidative phosphorylation cycles are mitigated. On the other hand, loss of cycling could contribute to impairments in mitochondrial function with age.

Both pathway analyses utilized here identified increased MESORs of steroid biosynthesis components in the aged sera condition. In particular, the genes identified in this pathway are associated with cholesterol biosynthesis. Since elderly individuals typically have lower levels of cholesterol biosynthesis and higher levels of circulating cholesterol (*Bertolotti et al., 2014*), we were surprised to see increased expression of biosynthesis transcripts. Perhaps, the cholesterol synthesis related RNA levels are high as a response to low levels of cholesterol synthesis proteins within the cell. Previous studies in cultured hepatocytes demonstrated that increased reactive oxygen species resulted in higher levels of transcripts associated with cholesterol biosynthesis (*Seo et al., 2019*). Given that deficits in oxidative phosphorylation are already implicated in these findings, it is possible that oxidative stress plays a role in the increase of cholesterol biosynthesis transcripts.

Although our findings are largely supported by the aging literature, the relatively small sample size of our study necessitates follow up studies to control for individual differences between subjects. We observed variations in luminescence and transcript traces across individuals and while we did not see changes that could account for the overall significant differences in transcript cycling between young and old subjects, we cannot exclude the possibility that factors other than aging contribute to these data.

Together, these findings indicate that at least some of the age-related changes in the cultured fibroblast circadian transcriptome are derived from signals circulating in the serum and not the age of the cells. This has profound implications for understanding and treating circadian disruption with age, and could also be relevant for other age-related pathology, given established links between circadian disruption and diseases of aging. Notably, many of the genes whose cycling is affected by old serum contribute to age-associated disorders.

## Materials and methods
### Clinical research study

This clinical research study enrolled apparent healthy participants from the volunteer pool maintained by the Institute for Translational Medicine and Therapeutics (ITMAT), University of Pennsylvania. The Institutional Review Board of the University of Pennsylvania (Federal wide

Assurance FWA00004028; IRB Registration: IORG0000029) approved the clinical study protocol (Penn IRB#832866). The study was registered on ClinicalTrials.gov with identifier NCT04086589. After obtaining informed consent from all volunteers, study assessments were conducted in the Center for Human Phenomic Science (Penn CHPS#3002) in accordance with relevant GCP guidelines and regulations. We originally intended on recruiting n=20 per group; however, patient recruitment was halted due to the COVID-19 pandemic. Participants met criteria for inclusion (in general good health, either 70–85 years of age for the elderly cohort or 20–35 years of age for the young cohort, and a wrist-actigraphy-based average TST (total sleep time)≥6 hours per night occurring between 22:00 – 08:00). Participants were excluded due to pregnancy or nursing, shift work (defined as recurring work between 22:00-05:00), a history of clinically significant obstructive sleep apnea, transmeridian travel across ≥2 time zones in the two weeks prior to study assessments and one week after, >2 drinks of alcohol per day, and use of illicit drugs. The subject-specific midsleep preferences were quantified using the Munich ChronoType Questionnaire (MCTQ) (*Roenneberg et al., 2003*). Blood collections were done from the median cubital vein via venipuncture using a 22 G butterfly needle (BD, Franklin, Lakes, NJ, USA). All blood draws occurred at 14:00, the same time as (*Pagani et al., 2011*). One subject returned for a repeat clinical assessment including biosampling to provide additional sample.

## Acquisition of accelerometry data streams

Participants wore a triaxial actigraph device (wGT3X-BT, ActiGraph, Pensacola, FL) on the non-dominant wrist. The devices were initialized using the following parameters: start date and time were synchronized with atomic server time without pre-defined stop date/time, at 60 Hz sampling rate for the three accelerometer axes, enabled for delay modus, steps, lux, inclinometer, and sleep while active. Raw data were downloaded from the device in AGD and GT3X file format in one second epochs using ActiLife software (version 6, ActiGraph, Pensacola, FL) and submitted for further analyses. For visualization and cosinor analysis, actigraphy data were aggregated into 1 min intervals by summing ActiGraph counts across each minute. Cosinor analyses of the data were adapted from the single component cosinor analysis reviewed by *Cornelissen, 2014*, as well as the cosine fit described by *Refinetti et al., 2007*. Briefly, the measurement times for the actigraphy data were recalculated as hours since midnight on the first day of measurement, within each participant's data. For each actigraphy variable and each participant, the lm() function in R (v4.2.0) was used to perform a cosinor fit with a fixed 24 hr period. The two cosinor coefficients from these fits were used to calculate participant-level amplitudes and circadian phases, while the intercepts provided MESOR estimates. The two-sided Wilcoxon rank sum exact test, as implemented by R's wilcox.test() function, was used to test for significant differences in amplitude and MESOR between the age groups. A two-sample Kuiper's test, as implemented by the kuiper_test(nboots = 10000) function from the twosamples R package (v2.0.0), was used to test for significant differences in circadian phase between the age groups.

## Acquisition of EKG data

The Zephyr BioPatch devices (Zephyr Technology, Annapolis, MD) were deployed as previously established (*Lahens et al., 2022*). All subjects included in this analysis wore the BioPatch for at least 24 hr. EKG recordings were analyzed by Kubios HRV Premium (ver. 3.5.0, Kubios Team, Kuopio, Finland) to report time-of-day-specific measures of cardiovascular function consisting of heart rate, RR intervals, sympathetic and parasympathetic nervous activity (SNS and PNS). Preprocessing in Kubios was set to automatic beat correction to remove artifacts and Smoothn priors for detrending. Cosinor fits and tests for differences in circadian parameters between age groups were performed as described for the actigraphy data. The circadian parameters MESOR and amplitude were tested for significant differences between the age groups by Wilcoxon rank sum exact test (two-sided), while phase was tested by Kuiper's two-sample test to account for the circular measurement.

## Cortisol measurements

Cortisol was measured in human serum by coated tube RIA (MP Bio, Solon OH) in duplicate. Tubes were counted on Perkin Elmer gamma counter and data reduced by STATLia software.

## Cell line

We used the BJ-5TA cell line from ATCC (CRL-4001). To prevent mycoplasma contamination Cells were treated with BM-Cyclin (Sigma 10799050001). Cells were maintained according to manufacturer instructions.

## Generation of stable BJ-5TA cell line expressing *BMAL1-dLuc-GFP* (*Ramanathan et al., 2012*)

Virus was generated and cells were infected as we've previously described (*Zhang et al., 2021*). Briefly, LentiX 293T cells (Clonetech) were transfected with Lipofectamine 3000 PLUS (Life Tech) using manufacturer instructions. The transfection included 18 µg of DNA per reaction and the plasmid (BMAL1-dLuc-GFP) to packaging vector (DVPR, Addgene) to envelope (VSV-G, Addgene) ratio was 10:1:0.5. Media was changed 24 hr post-transfection after the cells were checked using a fluorescence scope to make sure cells were >50% GFP positive. Supernatant with the virus was collected at 48 hr and 72 hr post transfection and spun down at 3000 RPM for 5 minutes (to eliminate any cells/debris). BJ-5TA cells were infected with fresh virus upon virus collection. Polybrene (Sigma-Aldrich, 10 mg/mL) was also added to aid with infection. Transduced BJ-5TA cells (>2000 cells) were sorted (FACSMelody, BD Biosciences) for high GFP expression. Once the cell line was established, blasticidin was added to the culture at 2 µg/ml. Due to fragility of the cell line in the presence of antibiotic, BJ-5TA *Bmal1-luciferase* cells with differing expression levels of GFP were sorted. The line with highest stable luminescence oscillation was used for all experiments reported here.

## Serum entrainment and bioluminescent recording

BJ-5TA *BMAL1-dLuc-GFP* cells in 24-well plates (~confluent) were washed (2 x) with DPBS and given serum free media for 24 hr. After the starvation, cells were given media with 10% human serum and 200 µM beetle luciferin potassium salt. Each well of cells was given media with the serum from a single patient. The cells were continuously monitored by LumiCycle luminometer (Actimetrics) for 4.5 days from the point of serum addition. LumiCycle raw data was exported using LumiCycle software (Actimetrics). The data were analyzed by BioDare2 (biodare2.ed.ac.uk) (*Zielinski et al., 2014*) using FFT NLLS with baseline detrending. Any replicates that were not cycling or had a period outside of the 20–28 hr range was excluded from analysis. Both the serum free media and serum added media used the recipe from *Ramanathan et al., 2012* at pH 7.4; however, the serum free media had no serum and the serum media had 10% human serum instead of FBS.

## Sample preparation, RNA extraction, and RNA sequencing

BJ-5TA *BMAL1-dLuc-GFP* cells in 24-well plates (~confluent) were washed (2 x) with DPBS and given serum free media (DMEM with Penn Strep) for 24 hr. After 24 hr cells were given media with human serum (DMEM, Penn Strep, 10% human serum). Serum starvation was staggered every 12 hr over 2 days to allow for samples to be collected on the same day. Upon sample collection wells were place on ice and rinsed with cold DPBS and then put in cold RLT buffer with 2-Mercaptoethanol (10µL/mL). Samples were frozen at –80 overnight and sent to Admera for RNA extraction and sequencing. Total RNA was extracted with RNeay mini kit (QIAGEN). Isolated RNA sample quality was assessed by High Sensitivity RNA Tapestation (Agilent Technologies Inc, California, USA) and quantified by Qubit 2.0 RNA HS assay (Thermo Fisher, Massachusetts, USA). Paramagnetic beads coupled with oligo d(T)25 are combined with total RNA to isolate poly(A)+transcripts based on NEBNext Poly(A) mRNA Magnetic Isolation Module manual (New England BioLabs Inc, Massachusetts, USA). Prior to first strand synthesis, samples are randomly primed (5′ d(N6) 3′ [N=A,C,G,T]) and fragmented based on manufacturer's recommendations. The first strand is synthesized with the Protoscript II Reverse Transcriptase with a longer extension period, approximately 40 min at 42 °C. All remaining steps for library construction were used according to the NEBNext UltraTM II Non-Directional RNA Library Prep Kit for Illumina (New England BioLabs Inc, Massachusetts, USA). Final libraries quantity was assessed by Qubit 2.0 (Thermo Fisher, Massachusetts, USA) and quality was assessed by TapeStation D1000 ScreenTape (Agilent Technologies Inc, California, USA). Final library size was about 430 bp with an insert size of about 300 bp. Illumina 8-nt dual-indices were used. Equimolar pooling of libraries was performed based on QC values and sequenced on an Illumina NovaSeq S4 Illumina, California, USA with a read length configuration of 150 PE for 40 M PE reads per sample (20 M in each direction).

## RNA-seq and statistical analysis

Raw RNA-seq reads were aligned to the GRCh38 build of the human genome by STAR version 2.7.10 a (*Dobin et al., 2013*). The dataset contained an average of 19,265,220 paired-end non-stranded 150 bp reads mapping uniquely to genes, per sample. Data were normalized and quantified at both gene and exon-intron level, using a downsampling strategy implemented in PORT (Pipeline Of RNA-seq Transformations, available at https://github.com/itmat/Normalization, *Itmat, 2021*), version 0.8.5f-beta_hotfix1. Both STAR and PORT were provided with gene models from release 106 of the Ensembl annotation (*Yates et al., 2020*).

MESOR, amplitude, and phase estimates, as well as p-values for the difference in MESOR, amplitude, and phase, were calculated with CircaCompare (*Parsons et al., 2020*), version 0.1.1. Only rhythmic genes were taken into consideration in pathway and other analyses involving MESORs and phases. The criterion for rhythmicity was either based on (BH adjusted) p-values reported by Circa-Compare, or weighted BIC values, obtained by an approach similar to dryR (*Weger et al., 2021*). In the latter approach we fitted four models of rhythmicity, one modeling gene expression that is rhythmic in both cells treated with young or old sera, another modeling gene expression rhythmic in neither cells treated with young nor old sera, and two modeling gene expression rhythmic in either young or old sera-treated cells, respectively. The BIC values were calculated for the four models and were weighted to obtain numbers between 0 and 1. All methods were provided with $\log_{10}(1+\text{PORT}$ normalized count) values and were run on R, version 4.1.2, accessed through Python, version 3.9.9, via rpy2, an interface to R running embedded in a Python process, https://rpy2.github.io/, version 3.4.5. Additionally, we used Nitecap (*Brooks et al., 2022*) to visualize and explore circadian profiles of gene expression.

## STRING pathway analysis

We performed STRING (*Szklarczyk et al., 2019*) pathway analyses using STRING API version 11.5. Enrichment analyses were performed on the sets of genes rhythmic in both groups (weighted BIC > 0.75) with observed decrease in MESOR, increase in MESOR, advance in phase, and delay in phase according to CircaCompare q<0.05 criterion. We also performed analyses on the sets consisting of genes rhythmic only in young group (weighted BIC > 0.75), only in old group (weighted BIC > 0.75), and the set of genes rhythmic in both groups (weighted BIC > 0.75). Finally, two additional analyses were performed, on the sets of genes with observed decrease and increase of amplitude (CircaCompare q<0.05).

## Ingenuity pathway analysis

QIAGEN IPA (*Krämer et al., 2014*) was used to identify pathways enriched in various subsets of cycling genes. The following subsets of genes were identified using a combination of CircaCompare stats and BIC cutoffs: (1) Decreased MESOR in old sera (929 genes) – CircaCompare rhythmic q<0.05 in old, CircaCompare rhythmic q<0.05 in young, CircaCompare MESOR difference q<0.05, MESOR difference (old – young)<0. (2) Increased MESOR in old sera (515 genes) – same selection criteria as 'Decreased MESOR in old sera,' except MESOR difference (old – young)>0. (3) Phase advance in old sera (148 genes) - CircaCompare rhythmic q<0.05 in old, CircaCompare rhythmic q<0.05 in young, CircaCompare Phase difference q<0.05, phase difference (old – young)<0. (4) Phase delay in old sera (156 genes) – same selection criteria as 'Phase advance in old sera', except phase difference (old – young)>0. (5) Loss of rhythmicity in old sera (1519 genes) – weighted BIC > 0.75 for 'Rhythmic in Young but not Old' model. (6) Gain of rhythmicity in old sera (637 genes) – weighted BIC > 0.75 for 'Rhythmic in Old but not Young' model. (7) Rhythmic in old and young sera (1209 genes) – weighted BIC > 0.75 for 'Rhythmic in both Old and Young' model. Each of these gene subsets were processed separately with IPA's core analysis, using default parameters.

For visualization via heatmap, we perform three rounds of normalization within each gene. First, we mean-normalize the read counts within each age group and serum treatment group (A, B, C, D). This is to account for baseline differences between sera collected from the different subjects. Second, we collapse replicates at each timepoint by calculating their means. Third, we calculate Z-Scores across all timepoints, within each age group. Note, these normalization procedures are to aid with visualization of the data and were not used as part of the statistical analyses.

## Epigenetic landscape in silico deletion (LISA)

LISA analysis (*Qin et al., 2020*) was used to perform transcription factor binding analysis. LISA results were filtered by fibroblast. q<0.05 cut off was used within each of the 5 groups. Venn diagram was generated with https://bioinformatics.psb.ugent.be/webtools/Venn/.

## Acknowledgements

We wish to extend our sincere gratitude to the study volunteers. Ms. LaVenia Banas provided excellent logistical study support. We thank Dr. Andrew Liu for the BMAL1-luc plasmid. We thank the RIA Biomarker Core of the Penn Diabetes Research Center, P30-DK19525 for cortisol measurements. We thank Sara Bernardez-Noya and Rebecca Moore for input on analysis and data presentation. The project described was supported by the National Center for Research Resources and the National Center for Advancing Translational Sciences, National Institutes of Health, through Grant 5UL1TR001878 (AS) and National Heart, Blood, and Lung Institutes of Health, through Grant R00HL147212 (SLZ). CS is the Robert L McNeil Jr. Fellow in Translational Medicine and Therapeutics. AS is an investigator of the Howard Hughes Medical Institute. JES was supported by a training grant in Neuroscience (NIH T32-NS105607), an National Institutes of Health Diversity Supplement (NIH NS48471), and by a grant to the University of Pennsylvania from the Howard Hughes Medical Institute through the James H Gilliam Fellowship for Advanced Study program. The funders had no role in study design, data collection and analysis, decision to publish, or preparation of the manuscript.

## Additional information

### Competing interests

Amita Sehgal: Reviewing editor, *eLife*. The other authors declare that no competing interests exist.

### Funding

| Funder | Grant reference number | Author |
| --- | --- | --- |
| National Center for Research Resources | 5UL1TR001878 | Amita Sehgal |
| National Heart, Lung, and Blood Institute | R00HL147212 | Shirley L Zhang |
| National Institute of Neurological Disorders and Stroke | T32-NS105607 | Jessica E Schwarz |
| National Institute of Neurological Disorders and Stroke | NIH NS48471 | Jessica E Schwarz |
| Howard Hughes Medical Institute | Gilliam Fellowship | Jessica E Schwarz |
| Howard Hughes Medical Institute | Investigator | Amita Sehgal |

The funders had no role in study design, data collection and interpretation, or the decision to submit the work for publication.

### Author contributions

Jessica E Schwarz, Conceptualization, Formal analysis, Investigation, Methodology, Writing - original draft, Writing – review and editing; Antonijo Mrčela, Nicholas F Lahens, Formal analysis, Visualization, Methodology, Writing – review and editing; Yongjun Li, Cynthia Hsu, Carsten Skarke, Formal analysis, Methodology, Writing – review and editing; Gregory R Grant, Methodology, Writing – review and editing; Shirley L Zhang, Formal analysis, Supervision, Investigation, Methodology, Writing – review and editing; Amita Sehgal, Conceptualization, Supervision, Methodology, Writing - original draft, Writing – review and editing

### Author ORCIDs

Jessica E Schwarz ⓘ http://orcid.org/0000-0003-4831-225X
Nicholas F Lahens ⓘ https://orcid.org/0000-0002-3965-5624
Gregory R Grant ⓘ https://orcid.org/0000-0002-0139-7658
Shirley L Zhang ⓘ https://orcid.org/0000-0002-6672-2044
Amita Sehgal ⓘ https://orcid.org/0000-0001-7354-9641

### Ethics

This clinical research study enrolled apparent healthy participants from the volunteer pool maintained by the Institute for Translational Medicine and Therapeutics (ITMAT), University of Pennsylvania. The Institutional Review Board of the University of Pennsylvania (Federal wide Assurance FWA00004028; IRB Registration: IORG0000029) approved the clinical study protocol (Penn IRB#832866). The study was registered on ClinicalTrials.gov with identifier NCT04086589. After obtaining informed consent from all volunteers, study assessments were conducted in the Center for Human Phenomic Science (Penn CHPS#3002) in accordance with relevant GCP guidelines and regulations.
Clinical trial Registry: ClinicalTrials.gov. Registration ID: NCT04086589.

Reviewer #1 (Public review): https://doi.org/10.7554/eLife.88322.3.sa1
Author response https://doi.org/10.7554/eLife.88322.3.sa2

## Additional files

### Supplementary files

• Supplementary file 1. Demographic information for subjects in the study. Total sleep time (TST) was calculated from ActiLife 6 determined in-bed and out-of-bed times averaged across nights with available actigraphy data (≥7 nights). Old (N=8), Y-young (N=7). For the RNAseq we did a pilot experiment which had one young and one old (*Supplementary file 4*). The young sample didn't have enough serum left to include that individual in the actual experiment (n=4 per group), but the old subject did. So, one old subject was used in both figures and the young were different.

• Supplementary file 2. BioPatch EKG Data Inclusion Log. A log of which participants were included in our biopatch data based on equipment function.

• Supplementary file 3. Genes involved in the IPA cell cycle of chromosome replication pathway show decreased MESOR with age. Table of genes that are rhythmic in both conditions with a decreased MESOR with old serum treatment compared to young serum treatment by CircaCompare. All genes are involved in the cell cycle/DNA replication pathway.

• Supplementary file 4. Genes with increased MESOR in the IPA cholesterol biosynthesis pathway. Table of genes that are rhythmic in both conditions with a decreased MESOR with old serum treatment compared to young serum treatment by CircaCompare. All genes are involved in the cholesterol biosynthesis pathway.

• MDAR checklist

### Data availability

Sequencing data have been deposited in GEO under the accession code GSE270290. All other source data is provided in source data files labeled with the corresponding figure.

The following dataset was generated:

| Author(s) | Year | Dataset title | Dataset URL | Database and Identifier |
| --- | --- | --- | --- | --- |
| Schwarz JE, Mrčela A, Lahens NF, Li Y, Hsu CT, Grant G, Skarke C, Zhang SL, Sehgal A | 2024 | Evidence for a role of human blood-borne factors in mediating age-associated changes in molecular circadian rhythms | https://www.ncbi.nlm.nih.gov/geo/query/acc.cgi?acc=GSE270290 | NCBI Gene Expression Omnibus, GSE270290 |

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
