## [Editor Report · eLife Assessment]

The authors tested the hypothesis that age-dependent factors in human sera affect the core circadian clock or its outputs in cultured fibroblasts, and they provide **compelling** evidence that genes involved in the cell cycle and transcription/translation remain rhythmic in both conditions, genes associated with oxidative phosphorylation and Alzheimer's Disease lose rhythmicity in the aged condition, while the expression of cycling genes associated with cholesterol biosynthesis increase in the cells entrained with old serum. Together, the findings suggest that yet to be identified age-dependent blood-borne factors affect circadian rhythms in the periphery. The paper provides **fundamental** insights and a possible explanation for previous observations showing that circadian gene expression in peripheral tissues tends to dampen or phase-shift with age.

---

## [Referee Report · Reviewer #1 (Public review)]

Aging is associated with a number of physiologic changes including perturbed circadian rhythms. However, mechanisms by which rhythms are altered remain unknown. Here authors tested the hypothesis that age-dependent factors in the sera affect the core clock or outputs of the core clock in cultured fibroblasts. They find that both sera from young and old donors are equally potent at driving robust ~24h oscillations in gene expression, and report the surprising finding that the cyclic transcriptome after stimulation by young or old sera differs markedly. In particular, genes involved in the cell cycle and transcription/translation remain rhythmic in both conditions, while genes associated with oxidative phosphorylation and Alzheimer's Disease lose rhythmicity in the aged condition. Also, the expression of cycling genes associated with cholesterol biosynthesis increases in the cells entrained with old serum. Together, the findings suggest that age-dependent blood-borne factors, yet to be identified, affect circadian rhythms in the periphery. The most interesting aspect of the paper is that the data suggest that the same system (BJ-5TA), may significantly change its rhythmic transcriptome depending on how the cells are synchronized. While there is a succinct discussion point on this, it should be expanded and described whether there are parallels with previous works, as well as what would be possible mechanisms for such an effect.

Comments on revised version:

The authors have done a thorough revision of their manuscripts and provided convincing answers to all of my points. In particular, I applaud the authors for having added raw luminescence traces, and for providing Figure S5 on the amplitudes. Perhaps the authors could add a comment in the final text that the amplitudes are fairly low, 10^0.1 = 1.25 which means that the bulk of those genes has rhythms of at most 25%, which could reflect that the synchronization of the cells is partial.

---

## [Author Response]

The following is the authors’ response to the original reviews.

**Public Reviews:**

**Reviewer #1 (Public Review):**
Aging is associated with a number of physiologic changes including perturbed circadian rhythms. However, mechanisms by which rhythms are altered remain unknown. Here authors tested the hypothesis that age-dependent factors in the sera affect the core clock or outputs of the core clock in cultured fibroblasts. They find that both sera from young and old donors are equally potent at driving robust ~24h oscillations in gene expression, and report the surprising finding that the cyclic transcriptome after stimulation by young or old sera differs markedly. In particular, genes involved in the cell cycle and transcription/translation remain rhythmic in both conditions, while genes associated with oxidative phosphorylation and Alzheimer's Disease lose rhythmicity in the aged condition. Also, the expression of cycling genes associated with cholesterol biosynthesis increases in the cells entrained with old serum. Together, the findings suggest that age-dependent blood-borne factors, yet to be identified, affect circadian rhythms in the periphery. The most interesting aspect of the paper is that the data suggest that the same system (BJ-5TA), may significantly change its rhythmic transcriptome depending on how the cells are synchronized. While there is a succinct discussion point on this, it should be expanded and described whether there are parallels with previous works, as well as what would be possible mechanisms for such an effect.

We’ve expanded our discussion in the manuscript to discuss possible mechanisms and also how the genes/pathways implicated in our study relate to other aging literature.

Major points:Fig 1 and Table S1. Serum composition and levels of relevant blood-borne factors probably change in function of time. At what time of the day were the serum samples from the old and young groups collected? This important information should be provided in the text and added to Table S1.

We made sure to highlight the collection time in the abstract of the manuscript “We collected blood from apparently healthy young (age 25-30) and old (age 70-76) individuals at 14:001 and used the serum to synchronize cultured fibroblasts.” The time of blood draw is also in sections of the paper (Intro and Methods). Since Table S1 is demographic information, we did not think that the blood draw time fit best there, but hopefully it is now clear in the text.

Fig 2A. Luminescence traces: the manuscript would greatly benefit from inclusion of raw luminescence traces.

Raw luminescence traces have been added to Figure S3 (S3A).

Fig 2. Of the many genes that change their rhythms after stimulation with young and old sera, what are the typical fold changes? For example, it would be useful to show histograms for the two groups. Does one group tend to have transcript rhythms of higher or lower fold changes?

We’ve presented these data in Figure S5. There are a few significant differences, but largely the groups are similar in terms of fold change.

Fig. 2 Gene expression. Also here, the presentation would benefit from showing a few key examples for different types of responses.

Sample traces of genes that gain rhythmicity, lose rhythmicity, phase shift, and change MESOR are now illustrated in Figure S6.

What was the rationale to use these cells over the more common U2OS cells? Are there similarities between the rhythmic transcriptomes of the BJ-5TA cells and that of U2OS cells or other human cells? This could easily be assessed using published datasets.

The original rationale to use BJ-5TA fibroblast cells was that we were aiming to build upon an observation found in a previous study2 which showed that circadian period changes with age in human fibroblasts. While our findings did not match theirs, we think an added benefit of using the BJ-5TA line is that unlike U2OS cells, it is not a carcinoma derived cell line. We’ve added this point in lines 98-101.

Our study finds many more rhythmic transcripts compared to the previous studies examining U2OS cells. This can be attributed to several factors including differences in methods, including the use of human serum in our study, cell type differences, or decoupling of rhythms in some cancer cells. While a comparison of BJ-5TA cells and U2OS cells could be interesting, a proper comparison requires investigation of many data sets, since any pair of BJ-5TA and U2OS data sets will most likely differ in some detail of experimental design or data processing pipeline, which could contribute to observed differences in rhythmic transcripts.

That being said, we compared clock reference genes (see Author response image 1) between BJ-5TA and U2OS cells, comparing circadian profiles obtained from our data with those available on CircaDB. These circadian profiles exhibit many similarities and a few differences. The peak to trough ratios (amplitudes) are quite similar for ARNTL, NR1D1, NR1D2, PER2, PER3, and are about 25% lower for CRY1 and somewhat higher for TEF (about 15%) in our data. We find that the MESORS are generally similar with the exception of NR1D1 which is much lower and NR1D2 which is much higher in our data.

**Author response image 1. sa2fig1:** BJ-5TA and U2OS Cells Exhibit Similar Profiles of Circadian Gene Transcription. We compared the transcriptomic profiles of the BJ-5TA cells in young and old serum (left) to the U2OS transcriptomic data (right) available on CircaDB, a database containing profiles of several circadian reference genes in U2OS cells. This figure suggests that circadian profiles of these genes exhibit many similarities. We find that the peak to trough ratios (amplitudes) are similar for ARNTL, NR1D1, NR1D2, Per2, PER3, and that the MESORS are similar (with the exception of NR1D1 which is much lower and NR1D2 which is much higher in the BJ-5TA cells). We find that the amplitudes of CRY1 is ~25% lower and TEF is ~15% higher for the BJ5TA cells. The axis for plots on the left show counts divided by 3.5 in order to made MESORs of ARNTL similar to ease comparison.

For the rhythmic cell cycle genes, could this be the consequence of the serum which synchronizes also the cell cycle, or is it rather an effect of the circadian oscillator driving rhythms of cell cycle genes?

This is an interesting point. Given our previous data showing that the cell cycle gene cyclin D1 is regulated by clock transcription factors3, we believe the circadian oscillator drives, or at least contributes, to rhythms of cell cycle genes. However, the serum clearly makes a difference as we find that MESORs of cell cycle genes decrease with aged serum. This is consistent with the decreased proliferation previously observed in aged human tissue4.

While the reduction of rhythmicity in the old serum for oxidative phosphorylation transcripts is very interesting and fits with the general theme that metabolic function decreases with age, it is puzzling that the recipient cells are the same, but it is only the synchronization by the old and young serum that changes. Are the authors thus suggesting that decrease of metabolic rhythms is primarily a non cell-autonomous and systemic phenomenon? What would be a potential mechanism?

We are indeed suggesting this, although it is also possible that it is not cycling *per se*, but rather an overall inefficiency of oxidative phosphorylation that is conveyed by the serum. Relating other work in the field to our findings, we’ve added the following to our discussion: “Previous work in the field demonstrates that synchronization of the circadian clock in culture results in cycling of mitochondrial respiratory activity5,6 further underscoring the different effects of old serum, which does not support oscillations of oxidative phosphorylation associated transcripts. Age-dependent decrease in oxidative phosphorylation and increase in mitochondrial dysfunction7 has been seen in aged fibroblasts8 and contributes to age-related diseases9. We suggest that the age-related inefficiency of oxidative phosphorylation is conferred by serum signals to the cells such that oxidative phosphorylation cycles are mitigated. On the other hand, loss of cycling could contribute to impairments in mitochondrial function with age.”

The delayed shifts after aged serum for clock transcripts (but not for Bmal1) are interesting and indicate that there may be a decoupling of Bmal1 transcript levels from the other clock gene phases. How do the authors interpret this? could it be related to altered chronotypes in the elderly?

One possible explanation is that the delay of NPAS2, BMAL1’s binding partner, results in the delay of the transcription of clock controlled genes/negative arm genes. Since the RORs do not seem to be affected, Bmal is transcribed/translated as usual, but there isn’t enough NPAS2 to bind with BMAL1. In this case downstream genes are slower to transcribe causing the phase delay.

**Reviewer #2 (Public Review):**
Schwarz et al. have presented a study aiming to investigate whether circulating factors in sera of subjects are able to synchronize depending on age, circadian rhythms of fibroblast. The authors used human serum taken from either old (age 70-76) or young (age 25-30) individuals to synchronise cultured fibroblasts containing a clock gene promoter driven luciferase reporter, followed by RNA sequencing to investigate whole gene expression.This study has the potential to be very interesting, as evidence of circulating factors in sera that mediate peripheral rhythms has long been sought after. Moreover, the possibility that those factors are affected by age which could contribute to the weaken circadian rhythmicity observed with aging.Here, the authors concluded that both old and young sera are equally competent at driving robust 24 hour oscillations, in particular for clock genes, although the cycling behaviour and nature of different genes is altered between the two groups, which is attributed to the age of the individuals. This conclusion could however be influenced by individual variabilities within and between the two age groups. The groups are relatively small, only four individual two females and two males, per group. And in addition, factors such as food intake and exercise prior to blood drawn, or/and chronotype, known to affect systemic signals, are not taken into consideration. As seen in figure 4, traces from different individuals vary heavily in terms of their patterns, which is not addressed in the text. Only analysing the summary average curve of the entire group may be masking the true data. More focus should be attributed to investigating the effects of serum from each individual and observing common patterns. Additionally, there are many potential causes of variability, instead or in addition to age, that may be contributing to the variation both, between the groups and between individuals within groups. All of this should be addressed by the authors and commented appropriately in the text.

We are not aware of any specific feature distinguishing the subjects (other than age) that could account for the differences between old and young. The fact that we see significant differences between the two groups, even with the relatively small size of the groups, suggests strongly that these differences are largely due to age. Nevertheless, we acknowledge that individual variability can be a contributing factor. For instance, the change in phase of clock genes appears to be driven largely by two subjects. We have commented on this and individual differences, in general, in the discussion.

The authors also note in the introduction that rhythms in different peripheral tissues vary in different ways with age, however the entire study is performed on only fibroblast, classified as peripheral tissue by the authors. It would be very interesting to investigate if the observed changes in fibroblast are extended or not to other cell lines from diverse organ origin. This could provide information about whether circulating circadian synchronising factors could exert their function systemically or on specific tissues. At the very least, this hypothesis should be addressed within the discussion.

It is likely that factors circulating in serum act on several tissues, and so their effects are relatively broad. However, this would require extensive investigation of other tissues. We now discuss this in the manuscript.

In addition to the limitations indicated above I consider that the data of the study is an insufficiently analysis beyond the rhythmicity analysis. Results from the STRING and IPA analysis were merely descriptive and a more comprehensive bioinformatic analysis would provide additional information about potential molecular mechanism explaining the differential gene expression. For example, enrichment of transcription factors binding sites in those genes with different patters to pinpoint chromatin regulatory pathways.

We performed LinC similarity analysis (LISA) to study enrichment of transcription factor binding. Results are displayed in Fig 3B and in lines 157-168.

**Recommendations for the authors:**
The two reviewers and reviewing editor have agreed on the following recommendations for the authors:Major:(1) The bioinformatic analysis would benefit from a more thorough focus on variability between individuals. Specifically, the main conclusion of the manuscript could be significantly influenced by individual variabilities within and between the two age groups. This is of particular concern, as the groups are relatively small (four individual two females and two males, per group). In addition, the consideration of factors such as food intake and exercise prior to blood drawn, or/and chronotype, known to affect systemic signals should be more adequately explained. The lab is an experienced chronobiology lab, and thus we are confident that these factors had been thought of, but this needs to be better made clear.As seen in Figure 4, traces from different individuals vary heavily in terms of their patterns, which is not addressed in the text. Only analysing the summary average curve of the entire group may be masking the relevant data. Furthermore, there are many potential causes of variability, instead or in addition to age, that may be contributing to the variation both, between the groups and between individuals within groups. All of this should be addressed by the authors and commented appropriately in the text.

We are not aware of any specific feature distinguishing the subjects (other than age) that could account for the differences between old and young. The fact that we see significant differences between the two groups, even with the relatively small size of the groups, suggests strongly that these differences are largely due to age. Nevertheless, we acknowledge that individual variability can be a contributing factor. For instance, the change in phase of clock genes appears to be driven largely by two subjects. We have commented on this and individual differences, in general, in the discussion.

(2) The study would benefit from a more thorough analysis of the data beyond the rhythmicity analysis. Results from the STRING and IPA analysis were merely descriptive and a more comprehensive bioinformatic analysis would provide additional information about potential molecular mechanism explaining the differential gene expression. For example, enrichment of transcription factors binding sites in those genes with different patters to pinpoint chromatin regulatory pathways. This would provide additional value to the study, especially given the otherwise apparent lack of any mechanistic explanation.

We performed LinC similarity analysis (LISA) to study enrichment of transcription factor binding. Results are displayed in Fig 3B and in lines 157-168.

(3) There were some questions about the amplitude of the core circadian clock gene rhythms raised, which in other human cell types would be much higher. A comment on this matter and the provision of the raw luminescence traces for Fig 2A would be greatly beneficial.Addressing the same topic: what are the typical fold changes of the many genes that change their rhythms after stimulation with young and old sera? For example, it would be useful to show histograms for the two groups. Does one group tend to have transcript rhythms of higher or lower fold changes? The presentation of the manuscript would further benefit from showing a few key examples for different types of responses.

The average luminescence trace for each individual serum sample from Fig 2A has been added to Fig S3A.

We’ve presented the fold change data in Figure S5. There are a few significant differences, but largely the groups are similar in terms of fold change.

(4) There are several points that we recommend to consider to add to the discussion:What was the rationale to use these cells over the more common U2OS cells? Are there similarities between the rhythmic transcriptomes of the BJ-5TA cells and that of U2OS cells or other human cells? It should be relatively easy to address this point by assessing published datasets.

The original rationale to use BJ-5TA fibroblast cells was that we were aiming to build upon an observation found in a previous study2 which showed that circadian period changes with age in human fibroblasts. While our findings did not match theirs, we think an added benefit of using the BJ-5TA line is that unlike U2OS cells, it is not carcinoma derived cell line. We’ve added this point in lines 98-101.

Our study finds many more rhythmic transcripts compared to the previous studies examining U2OS cells. This can be attributed to several factors including differences in methods, including the use of human serum in our study, cell type differences, or decoupling of rhythms in some cancer cells. While a comparison of BJ-5TA cells and U2OS cells could be interesting, a proper comparison requires investigation of many data sets, since any pair of BJ-5TA and U2OS data sets will most likely differ in some detail of experimental design or data processing pipeline, which could contribute to observed differences in rhythmic transcripts.

That being said, we compared clock reference genes (see Author response image 1) between BJ-5TA and U2OS cells, comparing circadian profiles obtained from our data with those available on CircaDB. These circadian profiles exhibit many similarities and a few differences. The peak to trough ratios (amplitudes) are quite similar for ARNTL, NR1D1, NR1D2, PER2, PER3, and are about 25% lower for CRY1 and somewhat higher for TEF (about 15%) in our data. We find that the MESORS are generally similar with the exception of NR1D1 which is much lower and NR1D2 which is much higher in our data.

For the rhythmic cell cycle genes, could this be the consequence of the serum which synchronizes also the cell cycle, or is it rather an effect of the circadian oscillator driving rhythms of cell cycle genes?

This is an interesting point. Given our previous data showing that the cell cycle gene cyclin D1 is regulated by clock transcription factors3, we believe the circadian oscillator drives, or at least contributes to rhythms of cell cycle genes. However, the serum clearly makes a difference as we find that MESORs of cell cycle genes decrease with aged serum. This is consistent with the decreased proliferation previously observed in aged human tissue.

While the reduction of rhythmicity in the old serum for oxidative phosphorylation transcripts is very interesting and fits with the general theme that metabolic function decreases with age, it is puzzling that the recipient cells are the same, but it is only the synchronization by the old and young serum that changes. Are the authors thus suggesting that decrease of metabolic rhythms is primarily a non cell-autonomous and systemic phenomenon? What would be a potential mechanism?

It may not be the cycling *per se*, but rather an overall inefficiency of oxidative phosphorylation that is conveyed by the serum. Relating other work in the field to our findings, we’ve added the following to our discussion: “Previous work in the field demonstrates that synchronization of the circadian clock in culture results in cycling of mitochondrial respiratory activity5,6 further underscoring the different effects of old serum, which does not support oscillations of oxidative phosphorylation associated transcripts. Age-dependent decrease in oxidative phosphorylation and increase in mitochondrial dysfunction7 is seen also in aged fibroblasts8 and contributes to age-related diseases9. We suggest that the age-related inefficiency of oxidative phosphorylation is conferred by serum signals to the cells such that oxidative phosphorylation cycles are mitigated. On the other hand, loss of cycling could contribute to impairments in mitochondrial function with age.”

The delayed shifts after aged serum for clock transcripts (but not for Bmal1) are interesting and indicate that there may be a decoupling of Bmal1 transcript levels from the other clock gene phases. How do the authors interpret this? Could it be related to altered chronotypes in the elderly?

One possible explanation is that the delay of NPAS2, BMAL1’s binding partner, results in the delay of the transcription of clock controlled genes/negative arm genes. Since the RORs do not seem to be affected, Bmal is transcribed/translated as usual, but there isn’t enough NPAS2 to bind with BMAL1. In this case downstream genes are slower to transcribe causing the phase delay.

The discussion would also benefit from mentioning parallels and dissimiliarities with previous works, as well as what would be possible mechanisms for such an effect.

We’ve expanded our discussion in the manuscript to discuss possible mechanisms and also how the genes/pathways implicated in our study relate to other aging literature.

Minor:While time of serum collection is provided in the methods, it would be very useful to provide this information, along with the accompanying argumentation also at a more prominent position and to also add it to Table S1.

We made sure to highlight the collection time in the abstract of the manuscript “We collected blood from apparently healthy young (age 25-30) and old (age 70-76) individuals at 14:001 and used the serum to synchronize cultured fibroblasts.” The time of blood draw is also in sections of the paper (Intro and Methods). Since Table S1 is demographic information, we did not think that the blood draw time fit best there, but hopefully it is now clear in the text.

L73 EKG: define the abbreviation

We rewrote this paragraph, but defined the term where it is used the paper.

L77: transfected BJ-5TA fibroblasts. Mention in the text that these are stably transfected cells.

We added this to the text.

L88: Day 2 also revealed different phases of cyclic expression between young and old "groups" for a larger number of genes. Here it is only two donors, right?

Yes, we swapped out the word “groups” for “subjects”.

L115. MESORs of steroid biosynthesis genes, particularly those relating to cholesterol biosynthesis, were also increased in the old sera condition. This is quite interesting, can the authors speculate on the significance of this finding?

We’ve added discussion about this finding in the context of the literature in our discussion.

Fig 3. - FDRs are only listed for certain KEGG pathways, and gene counts for each pathway are also missing, which excludes some valuable context for drawing conclusions. Full tables of KEGG pathway enrichment outputs should be provided in supplementary materials. Input gene lists should also be uploaded as supplementary data files.

Both output and input files are included in this submission as additional files.

Line 322 - How many replicates were excluded in the end for each group? Providing this information would strengthen the claim that the ability of both old and young serum to drive 24h oscillations in fibroblasts is robust and not only individual.

Each serum was tested in triplicate in two individual runs of the experiment. Of the 15 serum samples, on one of the runs, a triplicate for each of two serum samples (one old, one young) was excluded. Given that only one technical replicate in one run of the experiment had to be excluded for one old and one young individual out of all the samples assayed, this supports the idea that young and old serum drive robust oscillations.

Line 373 - Should list which active interaction sources were used for analysis.

In this manuscript we used STRING (search tool for retrieval of interacting genes) analysis to broadly identify relevant pathways defined by different algorithms. From these data, we focused in particular on KEGG pathways.

**Reviewer #1 (Recommendations For The Authors):**
These comments are in addition to those provided above:Minor:L73 EKG: define the abbreviation

We rewrote this paragraph, but defined the term where it is used the paper.

L77: transfected BJ-5TA fibroblasts. Mention in the text that these are stably transfected cells.

We added this to the text.

L88: Day 2 also revealed different phases of cyclic expression between young and old "groups" for a larger number of genes. Here it is only two donor, right?

Yes, we swapped out the word “groups” for “subjects”.

L115. MESORs of steroid biosynthesis genes, particularly those relating to cholesterol biosynthesis, were also increased in the old sera condition. This is quite interesting, can the authors speculate on the significance of this finding?

We’ve added discussion about this finding in the context of the literature.

Fig.4 The fold change amplitude of the clock gene seems quite a bit lower than what is usually expected (for Nr1d1 it is usually 10 fold). The authors should provide an explanation and discuss this.

There are a variety of factors that contribute to the fold change amplitude of clock genes. First, the change in amplitude of clock genes is lower in vitro compared to in vivo samples. For example, in U2OS cell cultures the fold change in the cycling of Nr1d1 is only 2 fold and is not significantly different from the fold change we observe (as shown in the U2OS data from CircaDB plotted in Figure 1R). Second, the method of synchronization contributes to the strength of the rhythms. Serum synchronization is generally less effective at driving strong clock cycling than forskolin or dexamethasone although, as noted in the manuscript, it may promote the cycling of more genes. Lastly, rhythm amplitude is also dependent on the cell type in question so cell to cell variability also contributes to differences. However, overall, we do not find major differences in comparing the U2OS data and ours. Please note that the y-axis has a logarithmic scale.

What is the authors' strategy to identify which serum components that are responsible for the reported changes? This should be discussed.

In the future, we intend to analyze the serum factors using a combination of fractionation and either proteomics or metabolomics to identify relevant factors. We have added this to the discussion.

**Reviewer #2 (Recommendations For The Authors):**

Overall, the article is well-written but lacks some more rigorous data analysis as mentioned in the public review above. In addition to a more thorough analysis approach focusing much more heavily on individual variability, several other changes can be made to strengthen this study:Fig 3. - FDRs are only listed for certain KEGG pathways, and gene counts for each pathway are also missing, which excludes some valuable context for drawing conclusions. Full tables of KEGG pathway enrichment outputs should be provided in supplementary materials. Input gene lists should also be uploaded as supplementary data files.

Both output and input files are included in this submission as additional files.

Fig 1A. - Only n=5 participants were used for this analysis, explanation of the exclusion criteria for the other participants would be useful.

As Figure 1A is a schematic, we assume the reviewer is referring to Figure 1B. We’ve provided a flow chart of subject inclusion/exclusion in Figure S2.

Fig 2. - For circadian transcriptome analysis only n=4 participants were used - what criteria was used to exclude individuals, and why were only these individuals used in the end?

As patient recruitment was interrupted by COVID, we selected samples where we had sufficient serum to effectively carry out the RNA seq experiment and control for age and sex.

Line 322 - How many replicates were excluded in the end for each group? Providing this information would strengthen the claim that the ability of both old and young serum to drive 24h oscillations in fibroblasts is robust and not only individual.

Each serum was tested in triplicate in two individual runs of the experiment. Of the 15 serum samples, on one of the runs, a triplicate for each of two serum samples (one old, one young) was excluded. Given that only one technical replicate in one run of the experiment had to be excluded for one old and one young individual out of all the samples assayed, this supports the idea that young and old serum drive robust oscillations.

Line 373 - Should list which active interaction sources were used for analysis.

In this manuscript we used STRING (search tool for retrieval of interacting genes) analysis to identify relevant pathways. We do not present any STRING networks in the paper.

Line 68 - "These novel findings suggest that it may be possible to treat impaired circadian physiology and the associated disease risks by targeting blood borne factors." This is a completed overstatement that are cannot be sustained by the limited findings provided by the authors.

We’ve modified this statement to avoid overstating results.

(1) Pagani, L. *et al.* Serum factors in older individuals change cellular clock properties. *Proceedings of the National Academy of Sciences*
**108**, 7218–7223 (2011).

(2) Pagani, L. *et al.* Serum factors in older individuals change cellular clock properties. *Proc Natl Acad Sci U S A*
**108**, 7218–7223 (2011).

(3) Lee, Y. *et al.* G1/S cell cycle regulators mediate effects of circadian dysregulation on tumor growth and provide targets for timed anticancer treatment. *PLOS Biology*
**17**, e3000228 (2019).

(4) Tomasetti, C. *et al.* Cell division rates decrease with age, providing a potential explanation for the age-dependent deceleration in cancer incidence. *Proceedings of the National Academy of Sciences*
**116**, 20482–20488 (2019).

(5) Cela, O. *et al.* Clock genes-dependent acetylation of complex I sets rhythmic activity of mitochondrial OxPhos. *Biochimica et Biophysica Acta (BBA) - Molecular Cell Research*
**1863**, 596–606 (2016).

(6) Scrima, R. *et al.* Mitochondrial calcium drives clock gene-dependent activation of pyruvate dehydrogenase and of oxidative phosphorylation. *Biochimica et Biophysica Acta (BBA) - Molecular Cell Research*
**1867**, 118815 (2020).

(7) Lesnefsky, E. J. & Hoppel, C. L. Oxidative phosphorylation and aging. *Ageing Research Reviews*
**5**, 402–433 (2006).

(8) Greco, M. *et al.* Marked aging-related decline in efficiency of oxidative phosphorylation in human skin fibroblasts. *The FASEB Journal*
**17**, 1706–1708 (2003).

(9) Federico, A. *et al.* Mitochondria, oxidative stress and neurodegeneration. *Journal of the Neurological Sciences*
**322**, 254–262 (2012).